# A high-throughput newborn screening approach for SCID, SMA, and SCD combining multiplex qPCR and tandem mass spectrometry

Rafael Tesorero[1]*, Joachim Janda[1]*, Friederike Hörster[1], Patrik Feyh[1], Ulrike Mütze[1], Jana Hauke[1], Kathrin Schwarz[1], Joachim B. Kunz[2], Georg F. Hoffmann[1], Jürgen G. Okun[1]

1 Department of General Pediatrics, Division for Neuropediatrics and Metabolic Medicine, Center for Child and Adolescent Medicine and Dietmar Hopp Metabolic Center, University Hospital Heidelberg, Heidelberg, Germany, 2 Department of Pediatric Oncology, Hematology, Oncology and Immunology, University of Heidelberg, Heidelberg, Germany

☯ These authors contributed equally to this work.
* Rafael.Tesorero@med.uni-heidelberg.de (RT); Joachim.Janda@med.uni-heidelberg.de (JJ)

**Data Availability Statement:** All relevant data are within the paper and its Supporting Information files.

## Abstract

Early diagnosis of severe combined immunodeficiency (SCID), spinal muscular atrophy (SMA), and sickle cell disease (SCD) improves health outcomes by providing a specific treatment before the onset of symptoms. A high-throughput nucleic acid-based method in newborn screening (NBS) has been shown to be fast and cost-effective in the early detection of these diseases. Screening for SCD has been included in Germany's NBS Program since Fall 2021 and typically requires high-throughput NBS laboratories to adopt analytical platforms that are demanding in terms of instrumentation and personnel. Thus, we developed a combined approach applying a multiplexed quantitative real-time PCR (qPCR) assay for simultaneous SCID, SMA, and 1st-tier SCD screening, followed by a tandem mass spectrometry (MS/MS) assay for 2nd-tier SCD screening. DNA is extracted from a 3.2-mm dried blood spot from which we simultaneously quantify T-cell receptor excision circles for SCID screening, identify the homozygous *SMN1* exon 7 deletion for SMA screening, and determine the integrity of the DNA extraction through the quantification of a housekeeping gene. In our two-tier SCD screening strategy, our multiplex qPCR identifies samples carrying the *HBB*: c.20A>T allele that is coding for sickle cell hemoglobin (HbS). Subsequently, the 2nd tier MS/MS assay is used to distinguish heterozygous HbS/A carriers from samples of patients with homozygous or compound heterozygous SCD. Between July 2021 and March 2022, 96,015 samples were screened by applying the newly implemented assay. The screening revealed two positive SCID cases, while 14 newborns with SMA were detected. Concurrently, the qPCR assay registered HbS in 431 samples which were submitted to 2nd-tier SCD screening, resulting in 17 HbS/S, five HbS/C, and two HbS/β thalassemia patients. The results of our quadruplex qPCR assay demonstrate a cost-effective and fast approach for a combined screening of three diseases that benefit from nucleic-acid based methods in high-throughput NBS laboratories.

**Funding:** The newborn screening pilot study ("Expansion of Newborn Screening by an additional 28 target diseases") is generously supported by the Dietmar Hopp Foundation, St. Leon- Rot, Germany (2311220 and 1DH1911376 to G.F.H.). The funders had no role in study design, data collection and analysis, decision to publish, or preparation of the manuscript.

**Competing interests:** The authors have declared that no competing interests exist.

## Introduction

The first quantitative real-time PCR (qPCR)-based newborn screening (NBS) in Germany was implemented in August 2019 to detect severe combined immunodeficiency (SCID), a group of inherited primary immunodeficiencies with an incidence estimated to be between 1:30,000–1:50,000 and which are characterized by the absence or extremely low numbers of naïve T-cells [1]. Infants born with SCID typically are asymptomatic at birth, but if not diagnosed and treated early, the diseases turn fatal within the first year of life due to opportunistic infections. Curative human stem cell transplantation (HSCT), enzyme replacement or gene therapy within the first months of life substantially increase the survival rates of SCID patients [2]. Because SCID is characterized by low to undetectable levels of T-cells, it can be detected early by measuring thymic function through the quantification of T-cell receptors excision circles (TRECs) from peripheral blood [3]. TRECs are stable, non-replicative, extrachromosomal circular DNA byproducts generated during the T-cell receptor rearrangement that occur in about 70% of newly matured T-cells. They are elevated in healthy newborns, and decline with increasing age due to decreased thymic activity [4]. TRECs can be effectively and rapidly quantified by a qPCR assay from infant dried blood spots (DBS) used for NBS programs. The TREC assay was first implemented in 2008 in Wisconsin, USA as a method for SCID screening [5]. Since then, several countries have nationally or regionally established SCID screening in their NBS programs by means of qPCR, opening up the possibilities for genetic screening of other diseases without a biomarker.

Another disease recently added to the German NBS panel that benefits from early detection from nucleic-based methods is spinal muscular atrophy (SMA), an autosomal recessively inherited disorder characterized by the degeneration of alpha motor neurons in the spinal cord, which results in progressive proximal muscle weakness and atrophy [6]. With an incidence of 1:6,000 to 1:10,000, SMA is the most common inherited neurodegenerative disease and was the leading genetic cause of death in early childhood [7,8]. The severity of symptoms depends on the SMA type classification, which is based on age of onset and achieved motor function [9]. The survival motor neuron (SMN) protein is encoded by two genes, the main functional *SMN1* gene and the paralog *SMN2* gene. The *SMN1* gene produces a functional SMN protein, while the *SMN2* gene encodes an attenuated form of the SMN protein, of which only 10–20% is functional [10]. *SMN1* is the disease-determining gene, and in SMA patients the *SMN2* gene copy number determines the type of SMA and its phenotypic severity [11]. The major difference between the *SMN1* and *SMN2* genes is a C to T change in exon 7 (c.840C>T), which causes a splicing error by generating transcripts lacking exon 7, and thus the expression of a defective SMN protein [12]. This single nucleotide change can identify the homozygous *SMN1* exon 7 deletion present in approximately 95% of SMA cases, and therefore serves as the primary target for early detection of SMA in NBS by qPCR [13]. Recently, the combined screening of SMA and SCID in a single multiplex qPCR assay has been established through a single multiplex qPCR assay, thus saving time and resources [14]. Subsequently, an already established multiplex qPCR assay can be modified to include other target diseases.

Sickle cell disease (SCD) is a heterogenous group of inherited blood disorders characterized by the sickle hemoglobin (HbS) allele, either present in a homozygous (SCD-S/S) or compound heterozygous form with another pathogenic hemoglobin variant (e.g. SCD-S/C, SCD-S/β-thalassemia). HbS is caused by a nucleotide change in the sixth codon of the hemoglobin β chain gene (*HBB*: c.20A>T), which leads to a substitution of glutamine for valine (p. Glu7Val) [15]. Homozygosity for the HbS allele is the most severe and most common reason for SCD [16]. HbS is prone to polymerization and can cause erythrocytes to lyse and clog capillaries. Clinically, patients with SCD typically present at the age of three to four months

with painful, often life-threatening vaso-occlusive crises [17]. Despite being considered a disease most prevalent in African countries, parts of the Middle East, and the tropics, SCD has an estimated birth prevalence of approximately 1:4,000 in Germany [15,18,19]. Diagnosis relies on the phenotypic analysis of hemoglobin; commonly established analytical techniques used for SCD screening include capillary electrophoresis (CE), high-performance liquid chromatography (HPLC), iso-electric focusing, tandem mass spectrometry (MS/MS), or MALDI-TOF MS [20–22].

For high-throughput (e. g. ≥ 500 samples per day) NBS laboratories, the implementation of new target diseases may substantially impact their ability to operate economically, e. g. if new instruments are required or new assays cannot be integrated into existing analytical procedures. Considerable additional demands may also be placed on logistical processes, such as sample distribution, or on having the necessary personnel. The screening laboratories were confronted with these demands upon the implementation of SCD in the German screening panel. PCR, while used for genotyping and some prenatal diagnosis, has not been used for routine SCD screening [23]. Nevertheless, it has been proven that DNA-based methods can unambiguously detect the presence of HbS in DBS by specifically targeting the *HBB*: c.20A>T allele [24,25]. This raises the option to adapt such an approach as an initial screening so that it can be integrated into an existing multiplexed high-throughput qPCR environment to detect all specimens containing HbS alleles. This, however, also comprises samples of the HbS carrier state, HbS/A. Such individuals are typically asymptomatic and must not be reported due to the German Gene Diagnostics Law [26]. Therefore, a second method is mandatory within the screening process to differentiate HbS/A from the pathogenic SCD variants.

Here we introduce a novel two-tiered approach combining qPCR and MS/MS, in which high-throughput NBS for SCID, SMA, and the presence of the *HBB*: c.20A>T allele is performed by a multiplex qPCR assay. To distinguish the carrier state from specimens with SCD within the preselected HbS-containing samples, and for phenotypic differentiation, an MS/MS assay is used as a 2nd-tier method. In addition, we outline the workflow developed in combining both analytical platforms and the outcomes obtained with this methodological approach during a three-month pilot study and six months of routine screening.

## Material and methods

### DBS samples

DBS samples were taken from the NBS laboratory of Heidelberg University Hospital. Three months prior to implementation of SMA and SCD to the German regular NBS on October 1, 2021, screening for SMA and SCD was integrated into the Heidelberg NBS pilot study "NGS2025", which includes the additional screening of 28 disorders (DRKS-ID DRKS00025324; approved by the Heidelberg ethics committee: application no. S- 533/2015) [27]. In the pilot period, only samples from newborns participating in this study (i.e. written informed consent of at least one parent/caregiver) were screened (*n* = 26,779). Likewise, NBS specimens used during method development were obtained only from filter cards of individuals participating in this study. With the start of SMA and SCD screening as part of the regular NBS panel in Germany on October 1, 2021, the screening workflow was applied to all NBS samples sent to our laboratory from then on (October 1, 2021 to March 31, 2022: *n* = 69,236). Positive SCD and SMA control samples, which served as references, were of patients treated in Heidelberg University Hospital after informed consent.

For the multiplex-qPCR assay, a single 3.2-mm punch per specimen was collected into a 96-well 0.2 mL plate (Sarstedt, Germany) using a Panthera-Puncher™ (Perkin-Elmer, Massachusetts USA). DBS samples with concentrated Bovine RBC (Innovative Research, Inc.) were

spiked with synthetic dsDNA (gBlock™ Gene Fragments, IDT, Iowa USA) to be used as negative and positive quality controls (QCs 1–3, S1 Table). Each 96-well plate contained duplicates of each QC and two blank wells (no template control, NTC). Samples for the $2^{nd}$-tier SCD method were manually punched from the same Guthrie cards used for the primary screening.

Diagnostic specimens used during method development and as positive quality controls for MS/MS were from adult patients with confirmed diagnoses who had given informed consent and were provided by the Department of Pediatric Oncology, Hematology, and Immunology at Heidelberg University Hospital.

## Chemicals

Acetic acid ($CH_3COOH$, LC-MS grade), acetonitrile (ACN) and methanol (MeOH) were purchased from VWR (Darmstadt, Germany). Formic acid (HCOOH, ULC/MS) was purchased from Biosolve (Valkenswaard, The Netherlands). Newborn haemoglobinopathy screening kits were bought from SpotOn Clinical Diagnostics Ltd. (London, Great Britain). Deionized water ($H_2O_{DI}$) was taken from an in-house preparation plant and ultra-pure water ($H_2O_{mQ}$) was prepared in-house using a GenPure Pro purification system from Thermo (Dreieich, Germany).

## DNA extraction

We utilized a modified two-step DNA extraction protocol that can be performed in a 96-well plate without the need of a column purification or precipitation step. Each DBS sample was first washed with 100 µL of Extracta DBS (Quantabio, Massachusetts USA) and centrifuged for 5 min at 2,082 x $g$. The supernatant was discarded using a Platemaster® P220 (Gilson, Middleton, Wisconsin USA), followed by an elution in 50 µL of Extracta DBS. The DBS samples were lysed by incubating in a thermal cycler (Applied Biosystems, Massachusetts USA) at 98˚C for 20 min and then brought to 4˚C. The DNA concentrations were not measured, and the lysate was diluted with 100 µL nuclease-free water (Invitrogen, Massachusetts USA) chilled at 4˚C and briefly centrifuged to decrease inconsistencies between replicates.

## qPCR assay design

The TREC assay was first implemented and accredited for SCID screening in our laboratory in August 2018 and was based on previously described methods [28,29]. Upon implementation of the new diseases, we kept the same TREC assay probes and modified our protocol accordingly. The combined TREC and *SMN1* analysis and the allele-specific PCR for the HbS mutation have been previously described [14]. Some of the oligos for TREC, *SMN1*, *ACTB*, and *HBB*: c.20A>T allele (hereinafter referred to as HbS allele) were based on previously described versions or have been modified thereof (Table 1) [14,24,28,30]. All oligos were synthetized by Integrated DNA Technologies (IDT, Iowa USA). The oligos for SCID screening are specific for the signal-joint region (δRec-ψJα) of TREC, which is in itself specific for actively differentiating T-cells in the thymus. SMA screening is performed in accordance with the recommendation from the German Joint Federal Committee (Gemeinsamer Bundsausschuss, G-BA) [31]. The screening is based on the homologous *SMN1* exon 7 deletion (also known as 5q-linked SMA) assay by targeting the c.840C>T single nucleotide variant. To increase target specificity and stability, we used the locked nucleic acid (LNA™) base modifications, such as the Affinity Plus™ monomers provided by IDT, to design the *SMN1* probe. Additionally, we used the same principle from the *SMN1* assay for the $1^{st}$-tier SCD screening probe design. SCD screening is based on the presence of the HbS allele, which results from a nucleotide change in the sixth codon of *HBB*; we therefore used the LNA modifications specific for the HbS allele to

**Table 1. Sequences of the primers and probes with their respective fluorophore reporter, quencher and nucleotide modifications used in the qPCR screening assay.**

| Target | Oligos | Sequence (5'-3') |
|---|---|---|
| TREC | Fwd primer | TTTGTAAAGGTGCCCACTCCT |
| | Rev primer | GCCAGCTGCAGGGTTTAGG |
| | TREC probe | **FAM**- CGGTGATGC/**ZEN**/ATAGGCACCTGCACC -**3IABkFQ** |
| SMN | Fwd primer | AATGCTTTTTAACATCCATATAAAGCTATC |
| | Rev primer | GAATGTGAGCACCTTCCTTC |
| | *SMN1* probe* | **SUN**- AGGG+TT+T+<u>C</u>+AGA+CAA -**3IABkFQ** |
| | *SMN2* blocker* | AGGG+TT+T+<u>T</u>+AGA+CAA |
| *HBB*: c.20A>T | Fwd primer | CAACTGTGTTCACTAGCAACC |
| | Rev primer | CCCCACAGGGCAGTAACG |
| | HbS allele probe* | **Cy5.5**- CT+C+CT+G+<u>T</u>+GGAG -**3IAbRQSp** |
| | *HBB* blocker* | CT+C+CT+G+<u>A</u>+GGAG |
| *ACTB* | Fwd primer | ATTTCCCTCTCAGGCATGGA |
| | Rev primer | CACGTCACACTTCATGATGGA |
| | *ACTB* probe | **Cy5**- CCTGTGGCA/**TAO**/TCCACGAAACTACCTTC -**3IAbRQSp** |

*Bases with + denote locked nucleic acid (LNA™) modifications. Underlined bases indicate nucleotide change of interest for *SMN1* exon 7 (c.840C) and *HBB*: c.20A>T alleles.

differentiate it from the non-pathogenic allele. To monitor the integrity of the DBS and DNA extraction, *ACTB* (β-actin) was utilized as an internal control target.

## qPCR preparation

The qPCR assay was performed in a 384-well plate including a 1x PerfeCTa® Multiplex qPCR ToughMix® with Low Rox (Quantabio, Massachusetts USA) master mix. End concentrations were 200 nM of TREC primers, 225 nM of *HBB* primers, 125 nM of SMN primers, and 50 nM of *ACTB* primers. The fluorescent probes specific for the corresponding target sequences used were 150 nM of FAM labeled probe for TREC with double-quenchers, 90 nM of SUN labeled probe for *SMN1*, and 200 nM of Cy5.5 labeled probe for HbS allele. Finally, 75 nM of Cy5 labeled probe for *ACTB* were added. Two unlabeled LNA probes were designed for *SMN2* and *HBB* as previously described and added to the quadruplex mix at a final concentration of 50 nM [30].

The master mix was distributed in a 384-well plate with a PIPETMAX® Liquid Handling System (Gilson, Wisconsin USA). This step allowed us to accurately dispense 12 μL master mix into each required well in the plate, while our custom protocol allowed us to flexibly adapt to varying requirements, e. g. depending on the number of samples. A total of 3 μL template from the 96-well DNA plate was distributed in a fixed manner into a 384-well plate with a second PIPETMAX® Liquid Handling System. The qPCR reaction was carried out in a 384-well block in the QuantStudio™ 7 Flex System (Applied Biosystems, Massachusetts USA). Each qPCR run included an initial denaturation at 95°C for 5 min, followed by 40 cycles of 95°C for 10 s, 60°C for 20 s, and 68°C for 15 s. Data collection was set during the 60°C annealing step.

## Sample preparation for 2nd-tier SCD analysis

The sample preparation was performed according to the instructions given by the hemoglobinopathy screening kit's manufacturer with minor modifications. Since the kit is designed as a first-tier assay, six aliquots each were prepared after the initial thawing of a pair of the kit's reagents, five of which were immediately refrozen and stored at -80°C until use. One 3.2-mm

punch of a dry blood sample was placed into a well of a polypropylene microplate (F-bottom, 96 well), and 50 μL each of prepared reagents 1 (solution of an isotopically labelled HbS fragment) and 2 (buffered trypsine solution), both diluted in deionized water, were added. The plate was sealed with a removable foil (Biozym, Hessisch Oldendorf, Germany), and after a short centrifugation step (2 min at $385 \times g$), the mixture was incubated for 90 min at 37°C with gentle shaking. Afterwards, the seal was removed and the reaction stopped by adding 200 μL of 0.1% HCOOH in $ACN/H_2O_{mQ}$ (9:1, $v/v$) to each well. The samples were centrifuged again (5 min at $690 \times g$), before 25 μL of the resulting supernatant were diluted with 225 μL mobile phase B in a second microplate (U-bottom, 96 well). After sealing this plate with a pierceable foil and 5 min of gentle mixing, 5 μL of this solution were subjected to instrumental analysis.

For quality control purposes, five additional control samples (one each of chemical blank, filter blank, HbS/S, HbS/C, and HbS/A) were included in each sample series along with the quality controls included in the kit.

## Tandem-MS

Samples were analyzed using a flow-injection MS/MS (FIA-MS/MS) system consisting of an Acquity UPLC binary solvent manager, 2777C sample manager, and a Xevo TQD MS (all from Waters, Eschborn, Germany) operated with an electrospray ionization source in positive mode with the following settings: Capillary voltage: 3.0 kV, source temperature: 150°C, desolvation temperature: 500°C, desolvation gas flow: 800 L/h, and cone gas flow: 250 L/h. Mobile phases were 0.3% $CH_3COOH$ in 80% $H_2O_{mQ}$/20% MeOH ($v/v$; A) and 0.3% $CH_3COOH$ in 20% $H_2O_{mQ}$/80% MeOH ($v/v$; B). A flow rate of 80 μL/min with 100% B was kept constant during the first minute of analysis. Subsequently, flushing at 400 μL/min with 100% A was performed for 0.3 min, then with 100% B for 0.4 min, and finally equilibration at the initial flow rate for 0.3 min. Informative mass transitions were monitored during the first minute of the measurement using an MRM experiment, applying constant cone voltage (30 V) and dwell times (50 ms). The corresponding transitions and collision voltages were obtained beforehand by infusing discrete solutions of the target peptides into the MS and are summarized in S2 Table. TargetLynx software was used to calculate the ratios between the signals of peptide fragments derived from pathogenic hemoglobin variants and—in most cases -their corresponding wild-type analogs, as well as hemoglobin F and A. The methodical workflows are illustrated in S1 Fig.

## Results

### qPCR analyses

After analysis of 4,610 samples of healthy newborns, cutoffs were established on the resulting copy numbers: the TREC cutoff was based on the 2.5[th] percentile, while *SMN1* and *ACTB* cutoffs were based on the 99.55[th] percentiles. For TREC, *ACTB*, and *SMN1* quantification, a standard curve was obtained through a serial 10-fold dilution of known copy numbers of synthetic dsDNA (gBlock, IDT) of each target. The TREC cutoff was set to five copies per punch, while *ACTB* was set at 1,226 copies per punch. Samples at or below the TREC or *ACTB* cutoff were re-tested from two new DBS. Duplicates with *ACTB* below the cutoff were considered unsatisfactory for DNA extraction from DBS and repeated with a second sample. If *ACTB* results were satisfactory, but TREC was not amplified or below the cutoff, then a T-cell deficiency would be suspected and a recall initiated.

For SMA screening, the *SMN1* cutoff was established at 1,114 copies per punch. Any sample below the cutoff was repeated from two new DBS as previously described. Samples that failed to amplify *SMN1*, but had satisfactory *ACTB* values exceeding the defined cutoff, were

considered SMA positive and forwarded for diagnosis. For 1st-tier SCD screening, no cutoff was set for HbS allele amplification. Any sample with a positive HbS allele amplification signal, including both heterozygous and homozygous, was subjected to 2nd-tier MS/MS screening.

## Positive controls

The sensitivity and specificity of our multiplex qPCR assay were tested with a number of positive and negative controls obtained from patients with confirmed diagnoses. In total, samples of six SMA patients, three SCID patients and 61 different HbS carrying individuals were blindly measured. All controls were correctly identified by our multiplex assay as depicted in Fig 1. In a typical sample from a healthy individual, TREC and *SMN1* were detected, while the lack of a positive HbS allele amplification signal indicated the absence of HbS and therefore a non-carrier individual. In a SCID patient, TRECs were not detected. Correspondingly, the absence of *SMN1* indicated an SMA patient. An amplified HbS allele signal in the plot indicated the presence of the HbS mutation in the corresponding sample, which would subsequently be analyzed by 2nd-tier MS/MS-based screening for differentiation. In all cases, *ACTB* serves as an internal control to determine DNA integrity after extraction, which is indicated by the corresponding amplification. To determine the specificity of the HbS probe, we additionally tested hemoglobin C (HbC) containing samples devoid of HbS, i. e. an HbC/C confirmed sample and HbC/A confirmed carriers. As depicted in S3 Fig, our multiplex assay did not amplify any HbC specimens that did not contain the HbS allele (results obtained for the same samples applying the MS/MS assay are given in S4 Table). Although qPCR alone may be applied to distinguish between heterozygous HbS/A and homozygous HbS/S, this principle was not expanded upon in our screening approach since the relevant compound-heterozygous SCD variants would be missed and a method to differentiate carrier and disease state would be required nonetheless.

## Tandem-MS analyses

A commercial MS/MS-based hemoglobinopathy screening kit was selected as the method for differentiation because the analytical platform already existed in our laboratory and there were published reports on, e.g., the diagnostic sensitivity and specificity achieved with the kit [21,32]. As part of the implementation of the assay, tuning experiments were performed to evaluate manufacturer-recommended and previously published mass transitions for the target peptides within the scope of the kit. To compare their sensitivity with the mass spectrometer being used, solutions of the individual peptides supplied by the kit manufacturer were infused into the instrument separately in concentrations of 1.0 μg/mL each. To increase the potential identification points in the assay, two mass transitions were monitored for most targeted peptides, as summarized in S2 Table. When different mobile phases were tested during the tuning experiments, we found that signal intensities increased substantially when a MeOH-based eluent (MeOH / $H_2O_{mQ}$ (4:1) + 0.3% acetic acid) was used instead of the mobile phase proposed in the original protocol. This effect is shown in S2 Fig for the HbS beta T1 peptide, where the methanolic mobile phase led to a signal increase of over 350% for the parent and daughter ions compared to the proposed ACN-based mobile phase. Considering the opportunity of further diluting the crude extracts and thus minimizing the matrix load on the instrument, we opted for the MeOH-based mobile phase.

Typically, SCD screening by means of MS/MS is evaluated by calculating diagnostic ratios obtained by dividing Hb variant signals by corresponding wild type signals [21]. The diagnostic ratios used here are summarized in S3 Table. Due to the application as a 2nd-tier method, the focus was on distinguishing between sickle cell trait (HbS/A) and SCD variants. Therefore,

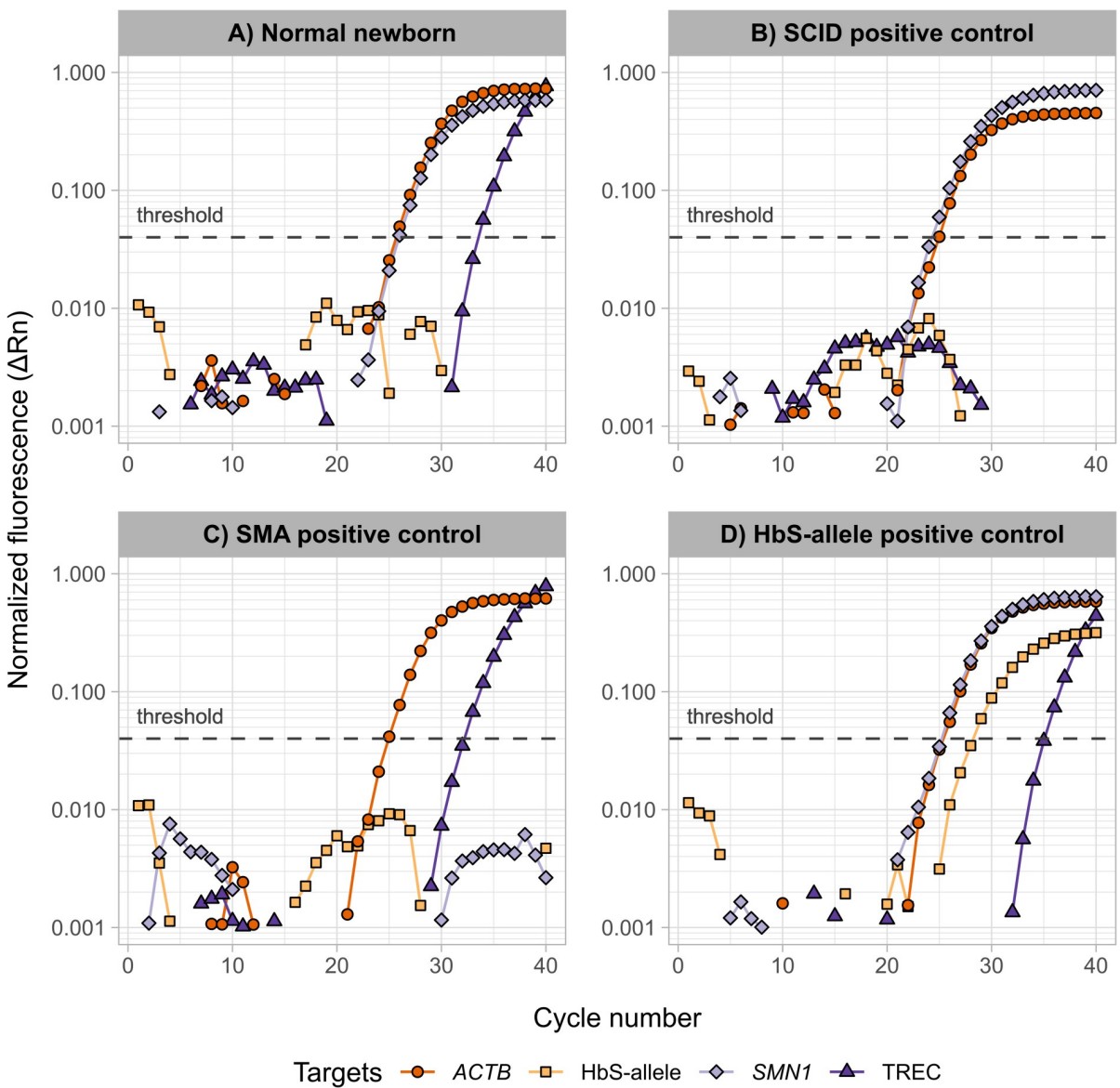

**Fig 1. Amplification plots for the fluorescent probes representing the four targets *ACTB*, HbS allele, *SMN1*, and TREC in samples of four different physiological conditions.** (A) A normal newborn, or screening negative, sample showing a normal amplification of *ACTB*, *SMN1*, and TREC, and no amplification of the HbS allele; (B) a reference SCID positive sample showing no TREC amplification; (C) a reference SMA positive sample with no *SMN1* amplification; (D) a sample showing HbS amplification indicating the allele presence. The figure was created using R (v. 4.1.2), ggplot2 (v. 3.3.5), dplyr (v. 1.0.7).

during the development phase, cutoffs for the two HbS/HbA ratios were determined by repeated analyses of samples from newborns with sickle cell trait ($n = 54$). After elimination of outliers, the 97.5[th] percentiles were calculated leading to cutoffs of 2.67 and 3.17 for HbS/HbA_1 and HbS/HbA_2, respectively. In our combined screening approach, wildtype samples are usually sorted out after the qPCR assay. However, for internal quality assurance and the option to categorize screen negative samples, threshold levels of 0.15 were set for both HbS/HbA ratios based on previously published data and results obtained using the MS/MS method on wildtype samples during method development. For the other diagnostic signal ratios (HbC/HbA, HbE/HbA, etc.), previously published action values were used [21].

An occasional problem encountered when calculating diagnostic ratios with the evaluation software commonly used in primary MS/MS-based screening for SCD (i.e., NeoLynx for Waters instruments) is division by zero. Given the opportunity in our two-tier approach to manually review and reintegrate the acquired signals of a comparably substantially reduced number of samples (compared to MS/MS applied as 1st-tier or general screening method) and thus minimize this issue, a TargetLynx-based evaluation method was set up, in which the HbA mass transitions were defined as "internal standards" for the respective ratios to be calculated.

## Screening results

**Pilot study.** During the three-month pilot study, the combined screening approach was employed in routine operation to test the methodology and improve the corresponding workflows. 26,779 NBS samples were analyzed in parallel to the existing NBS performed in the Heidelberg screening center. The newly implemented *SMN1* assay in our multiplex qPCR detected a total of four samples without a positive amplification of the *SMN1* gene. All samples showed a representative amplification of the β-actin reference gene and TREC quantification was satisfactory. These samples were repeated as duplicates in accordance with our screening algorithm, and the absence of the *SMN1* exon 7 was verified, with further confirmation of an SMA diagnosis in specialized accredited centers with further *SMN2* copy number determination. In the same period, no false negative or false positive *SMN1* exon 7 deletions were reported.

In the novel 1st tier NBS for SCD, qPCR detected HbS alleles in 120 samples, which were subsequently submitted to the MS/MS assay for further differentiation. Here, 78 samples were characterized as HbS/A and seven were SCD positive (six HbS/S and one HbS/C). These seven specimens were reported as conspicuous for the respective SCD variants and were further confirmed during patient follow-up. However, the remaining 35 samples did not contain HbS (i.e., most likely HbA/A) and the flagging for differentiation by MS/MS resulted from signal noise exceeding the defined threshold in the primary screening. In reproducibility tests, it was found that such samples could be identified in most cases by repetition of the qPCR analyses from two new DBS. Technically, in our screening approach, samples that may be erroneously flagged as suspicious for SCD due to high background noise would be identified during 2nd tier analysis in any case. Nevertheless, a control measurement by qPCR was introduced after the pilot phase for any suspicious sample to reduce the number of such cases and facilitate the routine workflow for the technical staff. In the MS/MS assay, two repeated injections were introduced whenever samples led to questionable results (e.g., one HbS/HbA ratio above and the other below the respective cutoff) or those that would lead to pathologic diagnoses in order to verify reproducibility and facilitate diagnostic decision-making.

**Routine screening.** The above-mentioned measures led to a workflow depicted in Fig 2, which was applied as part of regular screening from October 2021 onwards. Until March 31, 2022, 69,915 additional NBS samples were screened with our combined approach of the multiplex qPCR and 2nd-tier MS/MS assay for SCD screening. Two SCID cases, one non-SCID T-cell lymphopenia (TCL) case (CHARGE syndrome), and ten further SMA patients were detected while the two-tier approach for SCD screening resulted in 311 patient samples flagged as suspicious by qPCR and subsequently assayed by MS/MS. Of these, eleven samples were classified as homozygous HbS/S, five were compound-heterozygous HbS/C, two HbS/β-thalassemia (initially reported as HbS/S), and 285 samples HbS/A. All pathologic conditions found were confirmed during follow-up. One sample of a preterm infant had multiple elevated ratios due to low HbA levels and was recalled. The remaining eight samples gave HbS/HbA ratios < 0.15 and had been flagged due to signal noise exceeding the qPCR threshold.

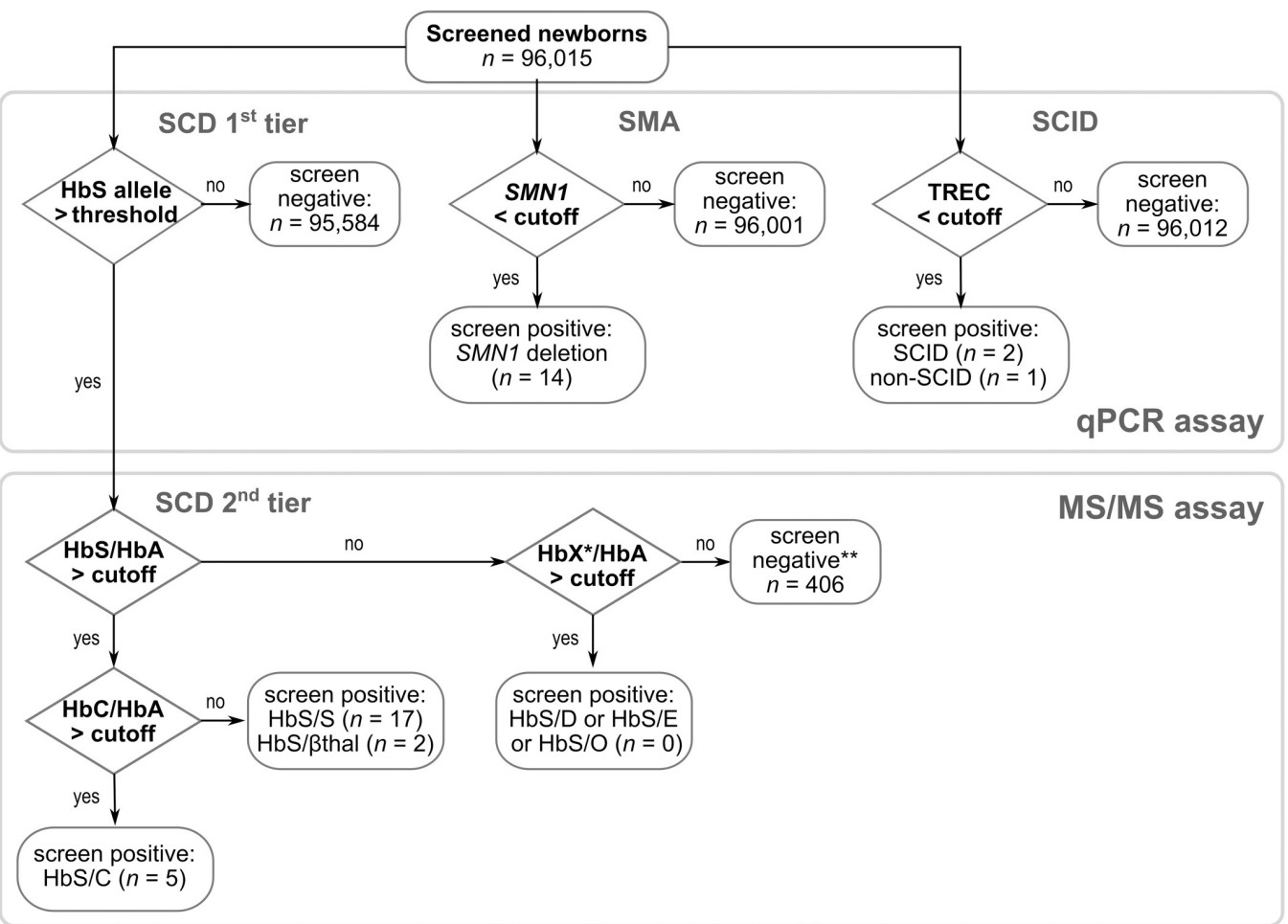

**Fig 2. The combined SCID, SMA and SCD screening.** Between July 1, 2021 and March 31, 2022, a total of 96,015 newborns were screened with a multiplex qPCR assay for SCID, SMA, and SCD. By the end of this period, a total of two SCID, one non-SCID syndromal lymphopenia,14 SMA, and 22 SCD patients were identified, * X = D$^{Punjab}$, E, or O$^{Arab}$; ** one preterm patient was recalled. The figure was created with InkScape (v. 1.2).

However, the introduction of the second PCR measurement reduced the share of such samples in the MS/MS differentiation method by over 90%.

## Discussion

A multiplexed qPCR assay was combined with an MS/MS assay enabling simultaneous high-throughput NBS for SCID, SMA, and SCD in a two-tiered setup. On peak days, more than 1,000 samples could be processed with the established multiplex qPCR workflow. To cope with such sample throughput, the qPCR analysis performing the primary screen requires a fast and efficient DNA extraction technique from DBS, an accurate liquid handling system, and a qPCR instrument capable of multi-channel detection in a 384-well format; this was realized in the qPCR method described. During a three-month pilot study and six months use in regular screening, 96,015 newborns were screened, which resulted in two SCID, one non-SCID T-cell lymphopenia, 14 SMA, and 24 SCD diagnoses, which were each confirmed after referral of the patients for follow-up to specialized and approved centers by the G-BA. After the nine-month screening period, the birth prevalence of SMA and SCD in southwestern Germany (approximately 1:6,857 and 1:4,000, respectively) is in accordance with former studies [7,8,19].

For SCD screening, particularly in high-throughput screening laboratories, qPCR can be a powerful tool to substantially decrease a large sample set to fewer HbS-containing specimens. In our case, its application reduced the sample size for the differentiation method through MS/MS by 99.6%. By implementing the LNA modifications in the HbS probe, just as it has been proven for the *SMN1* exon 7 deletion assay, it was possible to sort out samples devoid of the HbS allele quickly and reliably.

Moreover, the designed probe for the HbS allele proved to be highly specific, as it failed to hybridize with a HbC homozygous sample and several HbC heterozygous carriers. The genetic variant for HbC (*HBB*: c.19G>A) occurs at the same codon as for HbS (*HBB*: c.20A>T), but its presence did not affect our assay [33]. Nevertheless, lessons learned during the pilot phase helped to further optimize the workflow for regular SCD screening: Initially, samples screened positive for presence of HbS alleles by qPCR were automatically flagged for 2nd-tier analysis after their first measurement; in this stage, approximately 29% of these suspicious samples were flagged due to high background noise or artefacts effects only. After a second qPCR measurement for already flagged samples had been introduced into our workflow, this share could be substantially reduced to 2.6% in regular screening leading to smaller sample sequences in the 2nd-tier analyses. According to Guideline 025/016 of the German Society for Pediatric Oncology and Hematology, the confirmatory analysis performed as part of the follow-up of SCD patients should have been performed by the 28th day of life [34]. This requirement can be effortlessly met even with an additional qPCR and a weekly 2nd-tier analysis in NBS.

Two major pitfalls have been discussed before the implementation of SCD into the German NBS program: blood transfusion may cause a false-negative, and prematurity up to the 34th week of gestation may cause false-positive results [26,35]. Both confounding factors are frequently found within our screening population—during our study, a frequency of 1:405 transfused and 1:77 premature samples were screened. As the results of qPCR are inherently unaffected by these confounders, the application of a genetic method for SCD mass screening has the major advantage to reduce the number of recalls for such newborns compared to a biochemical screening method. Recalls due to blood transfusion are particularly critical, as a resampling eight weeks after the last transfusion is mandatory, and the risk for lost-to follow-up cases increases by age.

The SpotOn MS/MS assay for SCD screening is originally designed as a 1st-tier analysis and its application has been previously reported by multiple groups [21,32]. Here, it has been used as a 2nd-tier assay which required the preparation of aliquots from the original kit reagents. Considering the mass transitions for the isotopically labelled digestion control and results of the quality controls run in each batch of samples, currently no negative effects on the functionality of the kit due to this additional thawing/freezing step have been observed. As expected, the MS/MS assay successfully distinguished the HbS carrier state from SCD variants, so that only the latter were reported. Figures for diagnostic sensitivity (95.8–100%) and selectivity (98.7–100%) depending on SCD variants have been reported earlier for the kit's principal target conditions in NBS laboratories with corresponding cutoff values for variants-to-wildtype ratios (other than HbS/HbA) as used here [21]. In such considerations, the SCD variants that respond with the same diagnostic ratios, i.e., HbS/HbA for SCD-S/S, SCD-S/β° thalassemia, and SCD due to combined HbS and hereditary persistence of fetal hemoglobin are grouped together, as their classification by genotype can be equivocal [19]. In Germany, this classification is performed during follow-up in specialized hematology centers. Similarly, in our study, two individuals with a pattern indicating SCD due to homozygous HbS/S turned out to be SCD-S/β-thalassemia in the confirmatory diagnostics. However, the primary goal of NBS is to identify patients who are at risk for a particular disease, and this goal was fully achieved for all screen positives detected in our study. Nevertheless, further evaluation criteria (e.g., gestation-

dependent F/A ratios) are currently being tested in our laboratory with the aim of further improving the assignment of SCD genotypes.

The drastic reduction of samples from 1st to 2nd-tier analysis mentioned above also translates into a more cost-effective operation compared to a scenario in which the full number of incoming samples would be analyzed by MS/MS (or HPLC, or CE): In the latter case, at least three analytical systems would be necessary to perform the daily work (not to mention personnel, consumables, chemicals). In contrast, in our approach the daily work (qPCR) can be managed by the same staff who did the nucleic acid-based SCID screening before the inclusion of SMA and SCD screening. For the 2nd-tier analyses a single MS-system is sufficient, which is run once a week.

While the screening approach combining multiplexed qPCR and MS/MS is the most suitable for our laboratory, the entailed component of two-tiered SCD screening could be adapted depending on the respective surrounding conditions, e.g., as standalone approach without SCID and SMA screening or in combination with another differentiation method. For the latter case, any method that is capable of distinguishing heterozygous carriers and SCD variants could be applied. Instrumental platforms typically used for such purposes as mentioned in the introduction (e.g., CE, HPLC, or MS/MS) are usually complex and costly. However, in this respect, qPCR-based primary screening for HbS alleles and thus reducing the sample size to a group of specimens relevant to further investigations might be a promising approach to open SCD screening to less complicated and/or inexpensive methods, such as chip-based microelectrophoresis, aqueous multi-phase systems based on cell density measurements, lateral flow immunoassays, or even classical electrophoresis [23,36–38].

## Conclusions

Our study highlights the excellent multiplexing capability of the qPCR platform by integrating a primary screen for SCD into an already existing two-plex NBS assay for SCID and SMA. Furthermore, this extended method can easily process with the sample numbers that occur in a high-throughput NBS environment. Compared to other platforms used for SCD mass screening, this approach is efficient and far less demanding with regard to instrumental, personnel, or spatial requirements. The qPCR-based 1st-tier screening reliably detected samples containing HbS alleles, substantially reducing the sample amount for the 2nd-tier differentiation method. Moreover, due to the inherent sensitivity of qPCR, the method is largely unimpaired by typically challenging samples, such as those from preterm or transfused patients reducing their recall rate in SCD screening. In the 2nd-tier MS/MS method, a simple change of the mobile phase led to substantially increased signal intensities, enabling work with further diluted extracts, which may have a positive impact on maintenance cycles and instrument stability.

Although in the presented setup MS/MS is applied after the qPCR primary screen, any method appropriate to discriminate SCD phenotypes and heterozygous HbS carriers could be used as a 2nd-tier method, which may in turn provide an opportunity for SCD screening in less industrialized countries.

## Supporting information

**S1 Fig. Workflow representing the tasks for sample preparation and analysis.** (A) For the qPCR-based approach for SCID, SMA and 1st-tier SCD newborn screening. (B) For the MS/MS-based differentiation for 2nd-tier SCD newborn screening. This figure is accompanying the decision workflow depicted in Fig 2 of the main manuscript; prepared with Inkscape

(1.2.2).
(PDF)

**S2 Fig. Mass spectra for the HbS bT1 peptide (Parent Ion) acquired in MS1 scan and its y7-fragment (daughter Ion) acquired in MS2 scan.** All spectra were recorded in 'Multi Channel Analysis' mode Acquisition during 30 s with cycle time set to automatic and 2.0 s scan duration. The spectra were obtained by infusing solutions of the peptide (1.0 µg/mL) dissolved in an acetonitrile-based (solid line) and methanol-based (dotted) mobile phase.
(PDF)

**S3 Fig. Amplification plots for four exemplary HbC-containing specimens devoid of HbS (three HbC/A and one HbC/C).** The signal for the HbS allele is not amplified in any of the samples. The corresponding MS/MS results are shown in S4 Table.
(PDF)

**S1 Table. Quality controls (QCs) composition used in the qPCR assay.** Bovine RBC was 1:2 diluted with TE buffer pH 8.0 and different synthetic dsDNA (gBlocks, IDT) were used to spike the different QCs. RBC: Red blood cells.
(PDF)

**S2 Table. Targeted Hb-mutations, assigned mass transitions, and optimized MS parameters used in the 2$^{nd}$ tier instrumental analysis.**
(PDF)

**S3 Table. Diagnostic ratios used in the 2$^{nd}$ tier MS/MS method, the mass transitions for their calculation, and the action values applied.**
(PDF)

**S4 Table. Diagnostic ratios obtained with the MS/MS assay for the four HbC-containing specimens corresponding to the amplification plots depicted in S3 Fig.**
(PDF)

## Author Contributions

**Conceptualization:** Rafael Tesorero, Joachim Janda.

**Data curation:** Rafael Tesorero, Joachim Janda.

**Formal analysis:** Rafael Tesorero, Joachim Janda.

**Funding acquisition:** Georg F. Hoffmann.

**Investigation:** Rafael Tesorero, Joachim Janda, Jana Hauke, Kathrin Schwarz.

**Methodology:** Rafael Tesorero, Joachim Janda.

**Project administration:** Jürgen G. Okun.

**Resources:** Ulrike Mütze, Joachim B. Kunz.

**Software:** Patrik Feyh.

**Supervision:** Friederike Hörster, Georg F. Hoffmann, Jürgen G. Okun.

**Validation:** Rafael Tesorero, Joachim Janda, Friederike Hörster.

**Visualization:** Rafael Tesorero, Joachim Janda, Friederike Hörster, Ulrike Mütze, Jürgen G. Okun.

**Writing – original draft:** Rafael Tesorero, Joachim Janda.

**Writing – review & editing:** Rafael Tesorero, Joachim Janda, Friederike Hörster, Ulrike Mütze, Joachim B. Kunz, Jürgen G. Okun.

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
