## [Decision Letter · Decision Letter 0]

26 Dec 2022

PONE-D-22-32193A high-throughput newborn screening approach for SCID, SMA, and SCD combining multiplex qPCR and tandem mass spectrometryPLOS ONE

Dear Dr. Tesorero,

Thank you for submitting your manuscript to PLOS ONE. After careful consideration, we strongly think that your manuscript is well-written and addresses important topic. However, reviewers have several important comments for improving the quality of this manuscript. Therefore, we invite you to submit a revised version of the manuscript that addresses the points raised during the review process.

We look forward to receiving your revised manuscript.

Kind regards,

Elsayed Abdelkreem, MD, PhD

Academic Editor

PLOS ONE

Journal Requirements:

Reviewers' comments:

Reviewer's Responses to Questions

**Comments to the Author**

1. Is the manuscript technically sound, and do the data support the conclusions?

Reviewer #1: Yes

Reviewer #2: Yes

Reviewer #3: Yes

2. Has the statistical analysis been performed appropriately and rigorously? 

Reviewer #1: Yes

Reviewer #2: N/A

Reviewer #3: No

3. Have the authors made all data underlying the findings in their manuscript fully available?

Reviewer #1: Yes

Reviewer #2: Yes

Reviewer #3: Yes

4. Is the manuscript presented in an intelligible fashion and written in standard English?

Reviewer #1: Yes

Reviewer #2: Yes

Reviewer #3: Yes

5. Review Comments to the Author

Reviewer #1: “A high-throughput newborn screening approach for SCID, SMA, and SCD combining multiplex qPCR and tandem mass spectrometry“ which has been submitted by Rafael Tesorero, et al. to PLOS ONE, demonstrated the multiplexing capability of the qPCR platform by integrating a primary screen for SCD into an already existing duplex NBS assay for SCID and SMA. Furthermore, they extended the method to easily process with high sample numbers, realizing a high throughput NBS environment.

The authors described state-of-the-art techniques applicable to newborn screening. Many readers (including me!) can learn many things about advances in qPCR primers and probes, as well as sample preparation ingenuity for tandem MS.

Minor issues

(1) The authors did not detail the strength of their high-throughput analysis platform in the Discussion section. They only say something about that in the conclusion paragraph. However, this part should also be highlighted in the Discussion section.

If the Discussion section included the explanation about the high-throughput analysis technology beginning with a sentence such as "For high-throughput analysis, an efficient but easy DNA extraction technique from DBS and simultaneous qPCR measurement of a large number of samples are necessary", then, it would be good for anyone to understand the greatness of the authors’ work. The authors might think that the DNA extraction technique from DBS has been invented by somebody else, and qPCR machine for a large number of samples has already existed, thus they are not qualified to boast of their system. However, I think that the authors could boast of their idea of combination of a good DNA extraction technique and a good qPCR machine.

(2) The authors explained tandem mass spectrometry analysis as the second-tier assay in the Methods section and Results section, and argued its usefulness in the Discussion section.

I learned much knowledge on SCD and its screening with tandem mass spectrometry from their description in this article. Even so, I still have two questions.

①　I am wondering why the qPCR-screening-positive infants are not referred to an expert in hematology as soon as possible, without waiting the result of tandem mass spectrometry analysis.

②　I am also thinking another possibility that tandem mass spectrometry analysis as the second-tier assay can be replaced by another qPCR assay.

I would like to know how the authors answer to these questions.

(3) In the Methods section, the authors said “The combined TREC and SMN1 assays analysis was performed as previously described [14]”, citing the reference of Taylor et al (2015). However, the primers and probes of the authors were different from those of Taylor et al. In addition, as for SMA screening, the targeting SMN1-specific sequences were different between the authors and Taylor et al.

I think that the authors may have many reasons not to do as Taylor et al. did.

In addition, I am wondering whether the citation of Taylor et al. is proper here.

(4) I also would like to know why the authors use the dual quenchers in TREC and ACTB probes.

Are the dual quenchers essential for multiplexed qPCR? If the authors discussed these points in the Discussion section, it would be very helpful for readers (including me!).

(5) The authors said in the introduction section, “This single nucleotide change can identify the homozygous SMN1 exon 7 deletion present in approximately 95% of SMA cases, and therefore serves as the primary target for early detection of SMA in NBS by qPCR [13]. (lines 73-75)” I am afraid the content of the cited paper (Kubo et al. 2015) was not suitable here.

Reviewer #2: The article by Tesorero et al “A high throughput ……mass spectrometry” is well written and highlights the importance of multiplex PCR technology when it comes to population based high throughput screening. Here authors developed a combined approach for simultaneous SCID, SMA, and 1st-tier SCD screening followed by MS/MS 2nd tier SCD screening. Although adding SCD screening to the SCID/SMA multiplex assay is an unconventional approach since it is not adopted by newborn screening lab to my knowledge at least in the United States. Multiplex assay for SCID and SMA newborn screening is widely known to the newborn community while thin-layer isoelectric focusing (IEF) and high-performance liquid chromatography (HPLC) are routine technology most of newborn screening center or program use to perform SCD screening as first tier and second tier respectively. This first tier and second tier test provide SCD screen results with high sensitivity and specificity with some exception (transfused and premature samples). The results presented in this article are provided enough evidence through implementing this multiplex assay in German NBS program and implementing the MS/MS for 2nd tier SCD screening.

The question is whether the approach uses by the German NBS program can become benchmark for other NBS program? There is some sensitivity issue in the multiplex, and also in MS/MS approach as authors noted especially for SCD where initially high background noises prompted specimens for 2nd Tier MS/MS. Although authors able to reduce those false positive to minimum number, sensitivity and specificity will remain questionable for SCD in both 1st and 2nd tier. Also, whether the modification described in the 2nd Tier screening for accuracy to detect all variants related to SCD is well enough to detect all the hemoglobin variants related to SCD when compared to other technology, and whether this approach is cost effective? Authors should discuss more to these points in the article to make clear for the viewers and NBS community.

Here are more comments for authors to address:

1. Authors indicate that “control measurement was introduced to reduce the number SCD positive in 1st tier screening. Authors should include in the article what kind of approach was introduced for multiplex assay to reduce those number? Also, whether those secondary approach that was introduced was manual (person learning) or automated (machine learning) approach?

2. The authors indicates that during the pilot and regular screening combined they have 14 SMA. It is not cleared whether these 14 SMA are either a) just a Positive screen or b) confirmed SMA cases after diagnosis through SMN2 copy and clinical presentation.

Authors should make this statement very clear in the article because if all positive SMA screen doesn’t confirmed by diagnosis and clinical symptoms for SMA disease means some of the positive screenings are due to false positive cases.

Authors should also state somewhere whether their primer /probe designed for SMA screening can differentiate the SMA positive cases with the SMA hybrid cases. SMA hybrid transcript can come as a false positive sometimes depends on the binding of the primers and probes design.

3. Authors also suggest two confirmed SCID and one non-SCID T cell lymphopenia. Authors should write in the article how they come into these conclusions. Whether confirmed diagnosis was performed on those cases?

4. Authors have put lot of emphasis on SCD screening through multiplex with SMA and SCID, and 2nd tier MS/MS. This approach could become an opportunity for many NBS labs to adopt and implement along with already established SMA/SCID multiplex and performed the 2nd tier MS/MS (which is common technology for NBS lab). Authors also acclaimed that this approach can be use as model for point of care diagnostic for SCD as well in less industrialized countries. This is an important point given the disparity in health equity and equality in the NBS screening worldwide. Authors should highlight their view if their approach for SCD screening can reduce the burden in certain countries or races or populations where SCD cases are high.

5. Fig.2. Author indicated HbS allele> cut off whereas in the text (line 236) authors clearly wrote that “for 1st-tier SCD screening, no cutoff was set for HbS allele amplification”. Authors should explain why there are two different statements ( Fig2 vs Text for the same context).

Reviewer #3: In this study, two-tiered approach combining qPCR and MS/MS, has been developed for the diagnosis of SCID, SMA, and the presence of the HBB: c.20A>T allele via multiplex qPCR. The developed assay is promising and will enhance the detection of the target diseases in quite short period of time and with cost effective way. Nevertheless, the manuscript comprises some weaknesses which should be considered by the authors to take it to a better level. Some suggestions are as follow:

1. I would like to see a flow chart of a schematic diagram which summarizes the whole part of materials and methods and better illustrates the techniques implemented in this manuscript.

2. It was stated in line 253 that "data not shown", it will be better if the authors can show these data under the supporting information sections to avoid any misleading and show the bigger picture of their findings. This is crucial as the developed techniques can be implemented for diagnostic applications, and thus steak-holders and scientists would like to be aware about the full image of the authors findings.

3. In line 288, the authors mentioned that the cutoff ratio was set to 0.15 ratio! The selection of this ratio was based on what? Please elaborate more on this point.

4. In line 313: What do you mean by "second analysis"? Please be more specific to avoid any confusion for the reader.

5. I would like to see a separate conclusion section and not integrated under the discussion section.

6. The author should elaborate more about the potential of their developed assay for diagnostic applications and how it will be possible for this assay to be applied in hospitals and health care centers. This is should be under the conclusion section.

6. PLOS authors have the option to publish the peer review history of their article (what does this mean?). If published, this will include your full peer review and any attached files.

Reviewer #1: No

Reviewer #2: **Yes: **BINOD KUMAR

Reviewer #3: **Yes: **Prof. Hani A. Alhadrami

---

## [Author Response · Author response to Decision Letter 0]

8 Feb 2023

All comments and questions were addressed in the Reponse to reviewers document.

Reviewer #1:

1. The authors did not detail the strength of their high-throughput analysis platform in the Discussion section. They only say something about that in the conclusion paragraph. However, this part should also be highlighted in the Discussion section.

If the Discussion section included the explanation about the high-throughput analysis technology beginning with a sentence such as "For high-throughput analysis, an efficient but easy DNA extraction technique from DBS and simultaneous qPCR measurement of a large number of samples are necessary", then, it would be good for anyone to understand the greatness of the authors’ work. The authors might think that the DNA extraction technique from DBS has been invented by somebody else, and qPCR machine for a large number of samples has already existed, thus they are not qualified to boast of their system. However, I think that the authors could boast of their idea of combination of a good DNA extraction technique and a good qPCR machine.

Response:

Thank you for your feedback and generous comment. We have strengthened our discussion to emphasize the importance of a well noted DNA extraction and qPCR protocol for a high-throughput laboratory. The changes are as follows (lines 368-371): “To cope with such sample throughput, the qPCR analysis performing the primary screen requires a fast and efficient DNA extraction technique from DBS, an accurate liquid handling system, and a qPCR instrument capable of multi-channel detection in a 384-well format; this was realized in the qPCR method described.” 

2. The authors explained tandem mass spectrometry analysis as the second-tier assay in the Methods section and Results section, and argued its usefulness in the Discussion section.

I learned much knowledge on SCD and its screening with tandem mass spectrometry from their description in this article. Even so, I still have two questions.

2.1. I am wondering why the qPCR-screening-positive infants are not referred to an expert in hematology as soon as possible, without waiting the result of tandem mass spectrometry analysis.

Response:

The group of specimens preselected by qPCR contains not only SCD-affected but also those of heterozygous HbS/A individuals (carrier state). The latter must not be reported in Germany after the completed screening process due to the law on genetic diagnostics. Therefore, a differentiation to distinguish HbS/A (carrier state) from the disease state (SCD variants) is mandatory, which is in our case done by MS/MS. Since SCD mostly manifests between three to four months after birth (with the gradual exchange of fetal hemoglobin for adult hemoglobin), the short delay caused due to the two-tiered setup is acceptable (and is also considered in the hematologic guidelines for SCD follow-up in Germany). 

We had remarked the necessity for differentiation in our original manuscript in the introduction section, however, your question has indicated to us that this regulation needs to be expressed more clearly and we have therefore reworded the relevant passage 

 (lines 104- 107): “This, however, also comprises samples of the HbS carrier state, HbS/A. Such individuals are typically asymptomatic and must not be reported due to the German Gene Diagnostics Law [26]. Therefore, a second method is mandatory within the screening process to differentiate HbS/A from the pathogenic SCD variants.” 

In addition, we reworded the objective of our manuscript (lines 110-112): “To distinguish the carrier state from specimens with SCD within the preselected HbS-containing samples, and for phenotypic differentiation, an MS/MS assay is used as a 2nd-tier method.”

Furthermore, we follow the guidelines stablished by the German Society for Pediatric Oncology and Hematology for SCD screening. We added the following statement in the discussion section (lines 392-396): “According to Guideline 025/016 of the German Society for Pediatric Oncology and Hematology, the confirmatory analysis performed as part of the follow-up of SCD patients should have been performed by the 28th day of life [34]. This requirement can be effortlessly met even with an additional qPCR and a weekly 2nd-tier analysis in NBS.”

2.2. I am also thinking another possibility that tandem mass spectrometry analysis as the second-tier assay can be replaced by another qPCR assay.

Response:

You are correct, a second, more specific qPCR assay to determine SCD variants could be used. Theoretically, any method may be used as second tier method that fulfills two requirements: a) enabling to sort out heterozygous HbS/A specimens (carrier state, compare response to 2.1) and b) identify relevant variants of SCD. Typical established methods to achieve this are HPLC or capillary electrophoresis (or MS/MS), but a qPCR method applying probes designed to identify nucleotide sequences specific for the relevant HBB variants might be usable too. However, qPCR can be limited to the number of targets that can be simultaneously be detected based on technology availability.

We expanded our discussion section to highlight that other platforms could be used for the 2nd-tier analysis (lines 433-443): “While the screening approach combining multiplexed qPCR and MS/MS is the most suitable for our laboratory, the entailed component of two-tiered SCD screening could be adapted depending on the respective surrounding conditions, e.g., as standalone approach without SCID and SMA screening or in combination with another differentiation method. For the latter case, any method that is capable of distinguishing heterozygous carriers and SCD variants could be applied. Instrumental platforms typically used for such purposes as mentioned in the introduction (e.g., CE, HPLC, or MS/MS) are usually complex and costly. However, in this respect, qPCR-based primary screening for HbS alleles and thus reducing the sample size to a group of specimens relevant to further investigations might be a promising approach to open SCD screening to less complicated and/or inexpensive methods, such as chip-based microelectrophoresis, aqueous multi-phase systems based on cell density measurements, lateral flow immunoassays, or even classical electrophoresis [23, 36-38]”

3. In the Methods section, the authors said “The combined TREC and SMN1 assays analysis was performed as previously described [14]”, citing the reference of Taylor et al (2015). However, the primers and probes of the authors were different from those of Taylor et al. In addition, as for SMA screening, the targeting SMN1-specific sequences were different between the authors and Taylor et al.

I think that the authors may have many reasons not to do as Taylor et al. did.

In addition, I am wondering whether the citation of Taylor et al. is proper here.

Response:

Taylor et al. was the first peer reviewed publication on a combined SCID and SMA screening, and we used the LNA modifications and annealing temperature observations as a reference. Indeed, several of our oligos differ from the publications cited (line 165). We have kept the citation but changed the wording in our sentence to indicate how the aforementioned publications were used as a reference (lines 161-165): “The combined TREC and SMN1 analysis and the allele-specific PCR for the HbS mutation have been previously described [14]. Some of the oligos for TREC, SMN1, ACTB, and HBB: c.20A>T allele (hereinafter referred to as HbS allele) were based on previously described versions or have been modified thereof (Table 1) [14, 24, 28, 30].”

4. I also would like to know why the authors use the dual quenchers in TREC and ACTB probes.

Are the dual quenchers essential for multiplexed qPCR? If the authors discussed these points in the Discussion section, it would be very helpful for readers (including me!).

Response:

Before the implementation of our aforementioned quadruplex assay, we utilized the duplex TREC assay. During its optimization, we observed high background and decreased sensitivity with the FAM-labeled probe. IDT, the company we chose for the synthesis of our oligos, recommended that we utilize the double-quenched probes, which we did. This led to an observable improvement, followed by our TREC assay being accredited for SCID NBS by the German accreditation body (Deutsche Akkreditierungsstelle, DAkkS). Therefore, we kept the same modifications for future applications. Since the focus of our discussion is not the TREC assay, we decided not to mention nor focus on this aspect in the discussion section. However, we have included a statement under qPCR design to address the reviewer’s question (lines 159-161): “The TREC assay was first implemented and accredited for SCID screening in our laboratory in August 2018 and was based on previously described methods [28, 29]. Upon implementation of the new diseases, we kept the same TREC assay probes and modified our protocol accordingly.”

5. The authors said in the introduction section, “This single nucleotide change can identify the homozygous SMN1 exon 7 deletion present in approximately 95% of SMA cases, and therefore serves as the primary target for early detection of SMA in NBS by qPCR [13]. (lines 73-75)” I am afraid the content of the cited paper (Kubo et al. 2015) was not suitable here.

Response:

Noted and thank you for the observation, as Kubo et al. focused on intragenic mutations and hybrid SMN genes. We have replaced Kubo et al. with Arnold et al. 2015 as a more suitable citation (line 77).

Reviewer #2:

(…) The results presented in this article are provided enough evidence through implementing this multiplex assay in German NBS program and implementing the MS/MS for 2nd tier SCD screening. The question is whether the approach uses by the German NBS program can become benchmark for other NBS program?

1. There is some sensitivity issue in the multiplex, and also in MS/MS approach as authors noted especially for SCD where initially high background noises prompted specimens for 2nd Tier MS/MS. Although authors able to reduce those false positive to minimum number, sensitivity and specificity will remain questionable for SCD in both 1st and 2nd tier. Also, whether the modification described in the 2nd Tier screening for accuracy to detect all variants related to SCD is well enough to detect all the hemoglobin variants related to SCD when compared to other technology, and whether this approach is cost effective? Authors should discuss more to these points in the article to make clear for the viewers and NBS community.

Response:

Thank you very much for raising these important aspects. In our manuscript, we describe a screening approach and also address changes we adopted from the initial learning process during the pilot phase (added subsection, line 318). With our revision of the discussion section, we have taken up most of the aspects you mentioned. Nevertheless, we also add some thoughts about the points raised here:

• Our SCD screening approach has to be considered as a combination of qPCR and MS/MS methods which are required to be operated successional within a two-stage analytical process. Therefore, Wildtype samples that have passed the qPCR in 1st-tier, will always be detected in the 2nd-tier MS/MS. The numbers we gave on samples forwarded from 1st to 2nd tier due to “initially high background noises”, were to illustrate the learning process during the pilot phase and optimizations we implemented for routine screening later on. But these would not affect a “false positive” rate in a sense of a screening outcome. We have reworded the corresponding paragraphs to make this clearer (lines 386-392): “Nevertheless, lessons learned during the pilot phase helped to further optimize the workflow for regular SCD screening: Initially, samples screened positive for presence of HbS alleles by qPCR were automatically flagged for 2nd-tier analysis after their first measurement; in this stage, approximately 29% of these suspicious samples were flagged due to high background noise or artefacts effects only. After a second qPCR measurement for already flagged samples had been introduced into our workflow, this share could be substantially reduced to 2.6% in regular screening leading to smaller sample sequences in the 2nd-tier analyses.” 

• In Germany, the final target in SCD screening typically is the disease state “SCD”. A further differentiation is typically not required because screening results are regarded consumptive until they are confirmed in follow-up, which is obligatory performed in specialized centers (lines 418-419): “In Germany, this classification is performed during follow-up in specialized hematology centers.” 

Nevertheless, in our original manuscript, we already further differentiated outcome of the SCD types in our screening and added previously published data on diagnostic validation for MS/MS in the revision, but i) missing genotypes due to their rarity and ii) the fact that our screening approach is only running for a short time now makes it difficult to create a meaningful statement on selectivity and specificity. We reworded and added the following in the discussion section (lines 411-418): “As expected, the MS/MS assay successfully distinguished the HbS carrier state from SCD variants, so that only the latter were reported. Figures for diagnostic sensitivity (95.8-100%) and selectivity (98.7-100%) depending on SCD variants have been reported earlier for the kit’s principal target conditions in NBS laboratories with corresponding cutoff values for variants-to-wildtype ratios (other than HbS/HbA) as used here [21]. In such considerations, the SCD variants that respond with the same diagnostic ratios, i.e., HbS/HbA for SCD-S/S, SCD-S/β0 thalassemia, and SCD due to combined HbS and hereditary persistence of fetal hemoglobin are grouped together, as their classification by genotype can be equivocal [19].”

• Regarding the cost effectiveness, we would like to leave it at the qualitative statements that we added in the discussion, because this topic can quickly be subjected to political use in authorities within the German health care system (lines 426-432): “The drastic reduction of samples from 1st to 2nd-tier analysis mentioned above also translates into a more cost-effective operation compared to a scenario in which the full number of incoming samples would be analyzed by MS/MS (or HPLC, or CE): In the latter case, at least three analytical systems would be necessary to perform the daily work (not to mention personnel, consumables, chemicals). In contrast, in our approach the daily work (qPCR) can be managed by the same staff who did the nucleic acid-based SCID screening before the inclusion of SMA and SCD screening. For the 2nd-tier analyses a single MS-system is sufficient, which is run once a week.”

2. Authors indicate that a “control measurement was introduced to reduce the number SCD positive in 1st tier screening. Authors should include in the article what kind of approach was introduced for multiplex assay to reduce those number? Also, whether those secondary approach that was introduced was manual (person learning) or automated (machine learning) approach?

Response:

Before we get to the question about the control measurement, we would like to address that there are no “SCD positives” after 1st tier screening (please compare to question No 1). In our qPCR 1st-tier, specimens get flagged when “HbS alleles” are detected which does not necessarily indicate SCD. We reworded and expanded a section of our introduction to emphasize this point (lines 102-107): “This raises the option to adapt such an approach as an initial screening so that it can be integrated into an existing multiplexed high-throughput qPCR environment to detect all specimens containing HbS alleles. This, however, also comprises samples of the HbS carrier state, HbS/A. Such individuals are typically asymptomatic and must not be reported due to the German Gene Diagnostics Law [26]. Therefore, a second method is mandatory within the screening process to differentiate HbS/A from the pathogenic SCD variants.”

The term “control measurement” indicates here a repeat measurement of corresponding samples. This was the most straightforward way to check if the result for a specimen was reproducible. For most screen positives obtained in qPCR, such an approach would not be necessary because the corresponding analytical response is obvious. However, for the samples which were (likely) flagged due to high background noise, a repeated analysis was helpful. In routine operation, we did nevertheless include the repetition for all samples flagged as screen positives in 1st tier to facilitate the workflow for the technical staff.

To express this more clearly in the manuscript we reworded and expanded the following in the results section (lines 335-340): “In reproducibility tests, it was found that such samples could be identified in most cases by repetition of the qPCR analyses from two new DBS. Technically, in our screening approach, samples that may be erroneously flagged as suspicious for SCD due to high background noise would be identified during 2nd tier analysis in any case. Nevertheless, a control measurement by qPCR was introduced after the pilot phase for any suspicious sample to reduce the number of such cases and facilitate the routine workflow for the technical staff.”

3. The authors indicate that during the pilot and regular screening combined they have 14 SMA. It is not cleared whether these 14 SMA are either a) just a Positive screen or b) confirmed SMA cases after diagnosis through SMN2 copy and clinical presentation.

Authors should make this statement very clear in the article because if all positive SMA screen doesn’t confirmed by diagnosis and clinical symptoms for SMA disease means some of the positive screenings are due to false positive cases.

Authors should also state somewhere whether their primer /probe designed for SMA screening can differentiate the SMA positive cases with the SMA hybrid cases. SMA hybrid transcript can come as a false positive sometimes depends on the binding of the primers and probes design.

Response:

Thank you for pointing this out, as we did not explain enough how the cases were confirmed. All positive screened SMN1 exon7 deletion patients were confirmed on specialized centers outside the NBS and we also mentioned (lines 327) that as of today we have not reported false positive or false negative cases. To make it clearer to the reader, we reworded the following in the screening results section (lines 325-328): “…the absence of the SMN1 exon 7 was verified, with further confirmation of an SMA diagnosis in specialized accredited centers with further SMN2 copy number determination. In the same period, no false negative or false positive SMN1 exon 7 deletions were reported.”

We also refer in the introduction (lines 75-77) that SMA screening is based on the SMN1 exon7 deletion assay, which can detect up to 95% of the cases. Furthermore, SMA screening is performed with accordance the German law stablished by the G-BA (Gemeinsamer Bundesausschuss, Federal Joint Committee). We added in the materials and methods section the following (lines 167-170): “SMA screening is performed in accordance with the recommendation from the German Joint Federal Committee (Gemeinsamer Bundsausschuss, G-BA) [31]. The screening is based on the homologous SMN1 exon 7 deletion (also known as 5q-linked SMA) assay by targeting the c.840C>T single nucleotide variant.”

4. Authors also suggest two confirmed SCID and one non-SCID T cell lymphopenia. Authors should write in the article how they come into these conclusions. Whether confirmed diagnosis was performed on those cases?

Response:

The SCID and non-SCID T-cell lymphopenia cases were confirmed in specialized centers across Germany. The confirmation and diagnosis are done outside the NBS. We emphasize this point by rewording and adding the following in the discussion section (lines 373-375): “…which resulted in two SCID, one non-SCID T-cell lymphopenia, 14 SMA, and 24 SCD diagnoses, which were each confirmed after referral of the patients for follow-up to specialized and approved centers by the G-BA.” 

5. Authors have put lot of emphasis on SCD screening through multiplex with SMA and SCID, and 2nd tier MS/MS. This approach could become an opportunity for many NBS labs to adopt and implement along with already established SMA/SCID multiplex and performed the 2nd tier MS/MS (which is common technology for NBS lab). Authors also acclaimed that this approach can be use as model for point of care diagnostic for SCD as well in less industrialized countries. This is an important point given the disparity in health equity and equality in the NBS screening worldwide. Authors should highlight their view if their approach for SCD screening can reduce the burden in certain countries or races or populations where SCD cases are high.

Response:

In our closing sentence in the conclusions of the first manuscript version, we mentioned POC and the opportunity to adapt the SCD screening with a different 2nd-tier method for less industrialized countries. After internal discussions and reflection, we decided to remove the POC topic, because our method is intended for screening larger cohorts, which does not fit into the context of “bedside testing”.

Then again, we added a paragraph at the end of the discussion section, in which we discuss options, how qPCR might be combined with other analytical approaches to reduce the costs and possibly enable screening in less industrialized countries. We understand this as a perspective paragraph giving a thought-provoking statement for the SCD screening community (lines 433-443): “While the screening approach combining multiplexed qPCR and MS/MS is the most suitable for our laboratory, the entailed component of two-tiered SCD screening could be adapted depending on the respective surrounding conditions, e.g., as standalone approach without SCID and SMA screening or in combination with another differentiation method. For the latter case, any method that is capable of distinguishing heterozygous carriers and SCD variants could be applied. Instrumental platforms typically used for such purposes as mentioned in the introduction (e.g., CE, HPLC, or MS/MS) are usually complex and costly. However, in this respect, qPCR-based primary screening for HbS alleles and thus reducing the sample size to a group of specimens relevant to further investigations might be a promising approach to open SCD screening to less complicated and/or inexpensive methods, such as chip-based microelectrophoresis, aqueous multi-phase systems based on cell density measurements, lateral flow immunoassays, or even classical electrophoresis [23, 36-38].

6. Fig.2. Author indicated HbS allele> cut off whereas in the text (line 236) authors clearly wrote that “for 1st-tier SCD screening, no cutoff was set for HbS allele amplification”. Authors should explain why there are two different statements ( Fig2 vs Text for the same context).

Response:

Thank you for calling our attention to this error. The statement, that there was no cutoff established for 1st tier SCD screening is correct. Only the instrumental threshold, i.e. a binary yes/no result, was applied to flag specimens for presence of HbS alleles and necessity for 2nd tier analysis. We have modified figure 2 (see below) accordingly and changed the term “cutoff” for “threshold”.

Reviewer #3:

In this study, two-tiered approach combining qPCR and MS/MS, has been developed for the diagnosis of SCID, SMA, and the presence of the HBB: c.20A>T allele via multiplex qPCR. The developed assay is promising and will enhance the detection of the target diseases in quite short period of time and with cost effective way. Nevertheless, the manuscript comprises some weaknesses which should be considered by the authors to take it to a better level. Some suggestions are as follow:

1. I would like to see a flow chart of a schematic diagram which summarizes the whole part of materials and methods and better illustrates the techniques implemented in this manuscript.

Response:

The materials and methods section gives a detailed explanation of the techniques applied. Nevertheless, we have included S1 Figure (shown below) in the supporting information which summarizes our screening process. We added in line 233: “The methodical workflows are illustrated in S1 Figure.”

2. It was stated in line 253 that "data not shown", it will be better if the authors can show these data under the supporting information sections to avoid any misleading and show the bigger picture of their findings. This is crucial as the developed techniques can be implemented for diagnostic applications, and thus steak-holders and scientists would like to be aware about the full image of the authors findings.

Response:

Thank you for pointing out this – of course it is helpful to show such data and further support the statement about the specificity of the probe towards the HbS allele instead of the HbC sequence. We have prepared extra amplification plots acquired for corresponding samples (three HbC/A and one HbC/C) and added it in the SI (S3 Figure, see below). In addition, we have included a supporting table presenting the results for the MS/MS based diagnostic ratios of the same samples (S4 Table).

This section (lines 266-268) has been reworded to: “As depicted in S3 Figure, our multiplex assay did not amplify any HbC specimens that did not contain the HbS allele (results obtained for the same samples applying the MS/MS assay are given in S4 Table).

And reworded the following in the discussion section (lines 383-386): “Moreover, the designed probe for the HbS allele proved to be highly specific, as it failed to hybridize with a HbC homozygous sample and several HbC heterozygous carriers. The genetic variant for HbC (HBB: c.19G>A) occurs at the same codon as for HbS (HBB: c.20A>T), but its presence did not affect our assay [33].”

3. In line 288, the authors mentioned that the cutoff ratio was set to 0.15 ratio! The selection of this ratio was based on what? Please elaborate more on this point.

Response:

Thank you for raising this point, especially as the term "cutoff" can be misleading in the context presented here. Our 2nd-tier analysis aims for two major goals: 1) to distinguish between carrier (=HbS/A) and pathological states of SCD and 2) identify certain forms of SCD. Both require “cutoff” or action-values for the respective variant / wildtype ratios as presented in supporting information (S3 Table). 

A differentiation between wild type (HbA/A) and HbS/A with MS/MS is typically not necessary for us, because wildtype specimens are sorted out by qPCR. However, a value of 0.15 for the HbS/HbA ratios in MS/MS can be used to distinguish wild type and HbS/A. The values are based on results obtained during method development as well as previously published studies, in which MS/MS was used as standalone screening method for SCD. In our case, it could be regarded as a tool for quality assurance. We have reworded the section (lines 303-309) to express it more clearly: “In our combined screening approach, wildtype samples are usually sorted out after the qPCR assay. However, for internal quality assurance and the option to categorize screen negative samples, threshold levels of 0.15 were set for both HbS/HbA ratios based on previously published data and results obtained using the MS/MS method on wildtype samples during method development. For the other diagnostic signal ratios (HbC/HbA, HbE/HbA, etc.), previously published action values were used [21].”

4. In line 313: What do you mean by "second analysis"? Please be more specific to avoid any confusion for the reader.

Response:

Here, we were referring to the 2nd tier method. However, in response to Reviewer 2's question 2 about the control measurements, this paragraph has been edited and also this passage expressed more clearly. The changes are as follow (lines 335-338): “In reproducibility tests, it was found that such samples could be identified in most cases by repetition of the qPCR analyses from two new DBS. Technically, in our screening approach, samples that may be erroneously flagged as suspicious for SCD due to high background noise would be identified during 2nd tier analysis in any case.”

5. I would like to see a separate conclusion section and not integrated under the discussion section.

Response:

We have separated the conclusions into a special section (line 445). 

6. The author should elaborate more about the potential of their developed assay for diagnostic applications and how it will be possible for this assay to be applied in hospitals and health care centers. This is should be under the conclusion section.

Response:

The presented method is intended for NBS. In Germany, all diagnoses given in NBS are regarded as presumptive. Confirmation and definitive diagnosis are done outside the NBS in specialized centers. This is part of the legal framework in German health care system.

During the revision progress, we have set more emphasis on this regulation where necessary, but the fact that screening makes sense only for very large cohorts, we would prefer not to open the discussion to the potential use of our described method in hospitals or health care centers. Nevertheless, we picked up the point of altering the method for less industrialized countries and would, in this respect, kindly refer to question 5 of the second reviewer.

---

## [Decision Letter · Decision Letter 1]

1 Mar 2023

A high-throughput newborn screening approach for SCID, SMA, and SCD combining multiplex qPCR and tandem mass spectrometry

PONE-D-22-32193R1

Dear Dr. Tesorero,

We’re pleased to inform you that your manuscript has been judged scientifically suitable for publication and will be formally accepted for publication once it meets all outstanding technical requirements.

Kind regards,

Elsayed Abdelkreem, MD, PhD

Academic Editor

PLOS ONE

Additional Editor Comments (optional):

Reviewers' comments:

Reviewer's Responses to Questions

**Comments to the Author**

1. If the authors have adequately addressed your comments raised in a previous round of review and you feel that this manuscript is now acceptable for publication, you may indicate that here to bypass the “Comments to the Author” section, enter your conflict of interest statement in the “Confidential to Editor” section, and submit your "Accept" recommendation.

Reviewer #1: All comments have been addressed

Reviewer #2: All comments have been addressed

2. Is the manuscript technically sound, and do the data support the conclusions?

Reviewer #1: Yes

Reviewer #2: Yes

3. Has the statistical analysis been performed appropriately and rigorously? 

Reviewer #1: Yes

Reviewer #2: Yes

4. Have the authors made all data underlying the findings in their manuscript fully available?

Reviewer #1: Yes

Reviewer #2: Yes

5. Is the manuscript presented in an intelligible fashion and written in standard English?

Reviewer #1: Yes

Reviewer #2: Yes

6. Review Comments to the Author

Reviewer #1: I am satisfied with revisions that have been made by the authors. The authors have addressed all the critiques raised by this reviewer.

Reviewer #2: Thank you for addressing most of the queries related to the article. I believe with this improvement and additions, the MS is well read and may benefit to the NBS community.

7. PLOS authors have the option to publish the peer review history of their article (what does this mean?). If published, this will include your full peer review and any attached files.

Reviewer #1: No

Reviewer #2: **Yes: **Binod Kumar

---

## [Editor Report · Acceptance letter]

3 Mar 2023

PONE-D-22-32193R1 

A high-throughput newborn screening approach for SCID, SMA, and SCD combining multiplex qPCR and tandem mass spectrometry 

Dear Dr. Tesorero:

I'm pleased to inform you that your manuscript has been deemed suitable for publication in PLOS ONE. Congratulations! Your manuscript is now with our production department. 

Kind regards, 

on behalf of

Dr. Elsayed Abdelkreem 

Academic Editor

PLOS ONE